# Genome-Wide Identification and Phylogenetic and Expression Analyses of the *PLATZ* Gene Family in *Medicago sativa* L.

**DOI:** 10.3390/ijms24032388

**Published:** 2023-01-25

**Authors:** Xianyang Li, Fei He, Guoqing Zhao, Mingna Li, Ruicai Long, Junmei Kang, Qingchuan Yang, Lin Chen

**Affiliations:** 1Institute of Animal Science, Chinese Academy of Agricultural Sciences, Beijing 100193, China; 2College of Life Science and Technology, Harbin Normal University, Harbin 150025, China; 3Institute of Forage Crop Science, Ordos Academy of Agricultural and Animal Husbandry Sciences, Ordos 017000, China

**Keywords:** alfalfa, *MsPLATZ*, gene family, abiotic stress, genome-wide

## Abstract

The PLATZ family is a novel class of plant-specific zinc finger transcription factors with important roles in plant growth and development and abiotic stress responses. PLATZ members have been identified in many plants, including *Oryza sativa*, *Zea mays*, *Triticum aestivum*, *Fagopyrum tataricum*, and *Arabidopsis thaliana*; however, due to the complexity of the alfalfa reference genome, the members of the PLATZ gene family in alfalfa (*Medicago sativa* L.) have not been systematically identified and analyzed. In this study, 55 *Medicago sativa PLATZ* genes (*MsPLATZs*) were identified in the alfalfa “Xinjiangdaye” reference genome. Basic bioinformatic analysis was performed, including the characterization of sequence lengths, protein molecular weights, genomic positions, and conserved motifs. Expression analysis reveals that 7 *MsPLATZs* are tissue-specifically expressed, and 10 *MsPLATZs* are expressed in all examined tissues. The transcriptomic expression of these genes is obvious, indicating that these *MsPLATZ*s have different functions in the growth and development of alfalfa. Based on transcriptome data analysis and real-time quantitative PCR (RT-qPCR), we identified 22, 22, and 21 *MsPLATZ* genes that responded to salt, cold, and drought stress, respectively, with 20 *MsPLATZs* responding to all three stresses. This study lays a foundation for further exploring the functions of *MsPLATZs*, and provides ideas for the improvement of alfalfa varieties and germplasm innovation.

## 1. Introduction

Transcription factor is a protein that binds to specific DNA sequences, often containing one or more DNA-binding domains, and regulates the expression of target genes by identifying cis DNA sequences in the promoters of target genes [1,2]. Transcription factors play an important role in improving stress resistance in plants. Therefore, it is important to study the biological functions and regulatory mechanisms of important transcription factors under abiotic stress conditions. The typical DNA-binding regions in plant transcription factors include the AP2/EREBP domain [3], MYB domain [4], WRKY domain [5], zinc finger domain [6], and bZIP domain [7]. The PLATZ family is a novel class of plant-specific zinc finger transcription factors initially isolated from peas (first member named plant AT-rich sequence and zinc-binding protein 1, PLATZ 1), whose members contain two highly conserved protein zinc finger motifs (C-x_2_-H-x_11_-C-x_2_-C-x_(4–5)_-C-x_2_-C-x_(3–7)_-H-x_2_-H and C-x_2_-C-x_(10–11)_-C-x_3_-C) that show nonspecific binding to A/T-rich sequences and repress transcription [8]. PLATZ proteins play important roles in plant growth and development and abiotic stress responses.

Previous studies show that PLATZ proteins play important roles in multiple biological processes in plants, including regulating seed germination, leaf cell proliferation, and senescence. For example, the *Oryza sativa* PLATZ proteins can regulate the expression of genes involved in rice grain development. The *Oryza sativa GL6* (*SG6*) PLATZ protein knockout mutant gl6/sg6 shows grain length shortening and grain weight reduction, while after *GL6/SG6* overexpression, the grains become significantly larger and heavier, and plant height and plant fresh weight are significantly increased [9,10]. The protein encoded by the *Floury3* (*FL3*) gene in *Zea mays* also belongs to the PLATZ superfamily and is specifically expressed in endosperm cells and can affect endosperm development and nutrient storage [11]. A dominant *Arabidopsis thaliana* mutant of the PLATZ family transcription factor *ORESARA15* shows increases in the leaf life span and leaf size, suggesting that *ORE15* could regulate leaf growth [12]. In transcriptome comparison studies of normal villous (*FL*) and syngenic nonvillous (*Fl*) diploid cotton fibers, it was found that a PLATZ transcription factor was downregulated after 10 d of flowering in *Fl* compared with *FL*, suggesting that the PLATZ zinc finger transcription factor may play a role in the cotton fiber initiation and elongation stage [13]. A PLATZ protein plays an important role in the transition from primary growth to secondary growth during *Populus* stem development [14]. *PLATZ* gene expression is very different in mature and immature sugarcane tissues of sugarcane high-fiber genotypes, and it is speculated that PLATZ may be a transcriptional regulator of secondary cell wall synthesis [15].

Many studies show that PLATZ transcription factors are not only widely involved in plant growth and development, but can also respond to multiple abiotic stresses. *AtPLATZ1* plays an important role in seed dehydration tolerance. *AtPLATZ1* overexpression in wild-type *Arabidopsis thaliana* results in partial seed dehydration tolerance in vegetative tissues, contributing to its ability to improve low water utilization and survive in water-deficient environments, and *AtPLATZ1* regulates drought resistance in vegetative tissues [16]. The expression of *Zea mays PLATZ1* homologs is strongly induced by drought stress [14,15,16], indicating that PLATZ proteins are key factors regulating drought stress in *Zea mays* [17]. The heterologous expression of *GhPLATZ1* in *Arabidopsis thaliana* can inhibit ABA biosynthesis during seed development and regulate the *Arabidopsis thaliana* response to salt and osmotic stress during seed germination and seedling establishment. Therefore, it is speculated that *GhPLATZ1* may regulate osmotic stress and salt stress responses during cotton seed germination and seedling establishment [18].

Alfalfa (*Medicago sativa* L.) is a perennial legume plant widely planted worldwide as an economic forage plant [19]. It has the advantages of high adaptability, high protein content, and rich nutritional value [20,21]. Alfalfa has a long history of cultivation in China. With the continuous development of animal husbandry in China in recent years, the demand for alfalfa has also greatly increased, and research on alfalfa is intensifying [22]. Mining the excellent genes of *Medicago sativa* is of great significance to improve researching alfalfa at the theoretical level and to better guide the production of alfalfa. Therefore, the identification of PLATZ proteins in *Medicago sativa* is important based on the identified regulatory roles of PLATZ proteins in plant seed growth and development and the functional potential associated with plant stress resistance [9,12].

In this study, 55 PLATZ proteins were identified in the alfalfa genome. The chromosomal localizations, gene structures, conserved motifs, and cis-acting elements in the promoter regions of 55 *MsPLATZs* were comprehensively studied, and the evolutionary relationships of *Medicago sativa PLATZs* with those of three representative species, *Arabidopsis thaliana*, *Medicago truncatula*, and *Glycine max*, were analyzed. Real-time quantitative polymerase chain reaction (qRT-PCR) was used to determine the different tissue expression profiles of *MsPLATZs* in alfalfa. In addition, the expression profiles of *MsPLATZs* under three abiotic stresses (salt, drought, and cold stress) were also studied to determine the roles of *MsPLATZs* in different biological processes in alfalfa. These results provide valuable information for the identification of candidate *MsPLATZ* genes involved in various abiotic stress responses in alfalfa.

## 2. Results

### 2.1. Identification of PLATZ Genes in the Alfalfa Genome

A total of 55 *PLATZ* genes were identified in the genome of Chinese landraces (*Medicago sativa* L. XinJiangDaYe) on the basis of HMM and domain analysis [23]. Basic information about the members of the alfalfa PLATZ gene family is listed in Table 1 and Appendix A. As shown in Table 1, the full-length coding sequences (CDS), predicted protein products, molecular weights (Mw), and isoelectric points (pI) of the *MsPLATZ* genes are quite different, ranging from 156 (MsPLATZ1)–1890 (MsPLATZ17) bp, 52 (MsPLATZ1)–630 (MsPLATZ17) aa, 5.83 (MsPLATZ1)–72.48 (MsPLATZ17) kDa, and 4.69 (MsPLATZ10)–9.91 (MsPLAT37), respectively, with corresponding mean values of 654 bp, 218 aa, 24.82 kDa, and 8.92, respectively.

The 55 *PLATZ* genes are not evenly distributed on the 32 chromosomes in the alfalfa reference genome and are not distributed on either chr3 or chr6 (Figure 1). The greatest number of *MsPLATZ* genes (four genes) are located on chromosomes chr2.2, chr4.2, chr4.3, and chr8.1, while three genes are distributed on chromosomes chr2.3, chr2.4, chr4.1, and chr4.4. In chr1.1, chr1.2, chr1.3, chr1.4, chr2.1, chr5.2, chr5.3, chr5.4, chr8.2, and chr8.3, two genes are distributed on these 10 chromosomes. There is only one gene located on six chromosomes (chr5.1, chr7.1, chr7.2, chr7.3, chr7.4, and chr8.4). Finally, one gene is not distributed on the chromosome. We named these genes *MsPLATZ1-MsPLATZ54* based on their location on the chromosome, and a gene not on the chromosome, *MsPLZATZ55*.

### 2.2. Phylogenetic Analysis and Classification of MsPLATZ Proteins

To elucidate the evolutionary relationship between the MsPLATZ proteins and the model plant AtPLATZ proteins, a neighbor-joining tree containing 55 identified *MsPLATZs* and 12 *AtPLATZs* was constructed with MEGA software (Figure 2). The 67 PLATZ proteins were divided into five groups (I–V), and the MsPLATZ proteins were distributed in all five groups (I–V). Among the five groups, group V contains the most PLATZ proteins, with 21 members, including 15 *MsPLATZs* and 6 *AtPLATZs*, and group IV has the fewest *PLATZ* members, including 4 *MsPLAYTZs* and 1 *AtPLATZ*. Group II contains 3 *AtPLATZs* and 15 *MsPLATZs*, and the number of *MsPLATZs* is the same as that in group V. Group III contains only 10 *MsPLATZ* members. Group I has a total of 13 *PLATZ* members, including 11 *MsPLATZs* and 2 *AtPLATZs*. In addition, the phylogenetic tree of *MsPLATZs* was constructed based on the grouping in the phylogenetic tree and analyzed for gene structure and motif components among the groups (Figure 3A).

### 2.3. Gene structure and Conserved Motif Analysis of MsPLATZ Genes

To understand the structural composition of the *MsPLATZ* genes, we investigated the exon and intron structure of the *MsPLATZ* genes based on the *Medicago sativa* genomic DNA sequence (Figure 3B). The results show that the gene structure of the *MsPLATZ* genes presents some differences in the five groups, but also exhibits some similarity between the groups. The vast majority of genes (51 of 55, 92.73%) have five exons or fewer, with the greatest percentage containing four exons (40 of 55, 72.73%); only four genes, *MsPLATZ50/10/17/34*, have more than 5 exons, and *MsPLATZ34* has 13 exons. Among the genes in group I, *MsPLATZ1* has only one exon, and all other genes contain four exons. In group III, similar to group I, only *MsPLATZ10* has seven exons, and all other genes contain four exons. Group IV includes only four members, all of which contain four exons. Group V induces the gene with the largest number of exons, *MsPLATZ34*, with 13 exons. The number of exons varies among the members of group II, ranging from 1–8, but most of these genes contain four exons.

To identify the conserved motifs of the *MsPLATZs*, we performed an assay analysis with the MEME website. A total of 10 conserved motifs were detected in the *MsPLATZ* genes, which were designated Motifs 1–10 (Figure 3C). Motif 3, motif 7, motif 1, motif 2, and motif 6 represent 100%, 98.18%, 94.55%, 81.82%, and 80%, respectively, of the five most abundant pairs of the 10 motifs, indicating that these motifs are the most important components of MsPLATZ proteins. Motif 7 is not present in *MsPLATZ19*, indicating possible sequence loss in the course of evolution. The members within each group present similar gene structures and motifs, with some motifs only being present in specific groups, such as motif 10, which is only present in group IV, and motif 9, which is only present in group V. Thus, the *MsPLATZ* genes of the same group have similar gene structures and motif compositions, while *MsPLATZ* genes from different groups may have specific structures, indicating that the conservation and diversity of *MsPLATZ* function have evolved during long-term evolution.

### 2.4. Gene Duplication Events and Synteny Analysis of MsPLATZ Genes

To further investigate possible gene duplication events among the *MsPLATZ* genes, we mapped the predicted *MsPLATZs* using the genome database of *Medicago sativa* (XinJiang DaYe) [23]. The results show possible tandem duplication events among genes such as *MsPLATZ18/19/20* on chromosome chr2.4 (Figure 1). The analysis of segmental duplication events in the *MsPLATZ* family reveal a total of 94 pairs of fragment duplications, as found in *MsPLATZ21* and *MsPLATZ1*, located on different chromosomes (Figure 4). These results suggest that duplication events have been widely involved in the evolution of *MsPLATZs*.

Furthermore, to further understand the possible evolutionary events of the PLATZ gene family in different crops, collinearity maps of *Medicago sativa* with *Arabidopsis thaliana*, *Glycine max,* and *Medicago truncatula* were constructed. The *MsPLATZ* genes differ from those of the other three species. A total of 16 of the 55 *MsPLATZ* genes show collinearity with *Arabidopsis thaliana*, 42 with *Medicago truncatula*, and 43 with *Glycine max* (Figure 5). Among these genes, there are 20 pairs in *Arabidopsis thaliana*, 64 in *Medicago truncatula*, and 134 in *Glycine max* (Figure 5). The collinearity identified between *Medicago sativa*, *Medicago truncatula,* and *Glycine max* is significantly higher than that between *Medicago sativa* and *Arabidopsis thaliana*, indicating that the distribution of *PLATZ* genes is relatively conserved among the three legume species.

### 2.5. Expression Patterns of MsPLATZ Genes in Different Tissues of Medicago sativa

Genome-wide gene expression data for some tissues of alfalfa, such as roots, elongated stems, pre-elongated stems, leaves, flowers, and nodules, are publicly available. As shown in Figure 6A, only 23 of the 55 *MsPLATZ* genes are expressed in the six tissues studied, and other unexpressed genes may be expressed in other specific tissues or specifically expressed under certain biotic or abiotic stresses. The transcript abundance of the 23 expressed *MsPLATZ* genes also varies in the six tissues, indicating that the functions of each gene are also significantly different. A total of 7 out of 23 *MsPLATZ* genes are detected in only one tissue, indicating tissue expression specificity. The seven genes are *MsPLATZ47*, expressed in roots; *MsPLATZ31*, expressed in pre-elongated stems; and *MsPLATZ26/43/38/39/40*, expressed in flowers. The only gene expressed in three tissues is *MsPLATZ6*, which is expressed in flowers, leaves, and nodules. Two genes, *MsPLATZ1* and *MsPLATZ54*, are expressed in two tissues: *MsPLATZ1* is expressed in flowers and elongated stems, and *MsPLATZ54* is expressed in nodules and roots. *MsPLATZ32* is expressed in five of the tissues with the exception of leaves, and *MsPLATZ33/51* is expressed five of the tissues with the exception of pre-elongated stems. Ten genes are expressed in six tissues, including *MsPLATZ41/37/36*, and although they are expressed in six tissues, their expression is significantly higher in flowers than in other tissues. These results indicate significant functional differences between *MsPLATZ* genes, and suggest that they play important roles in regulating the growth and development of *M. sativa*.

### 2.6. Analysis of Promoter Cis-Acting Elements of MsPLATZ Genes

To further explore the functional potential of the *MsPLATZ* genes, we analyzed the cis-acting elements withing the 2 kb upstream region of the *MsPLATZ* start codon (ATG) using the PlantCARE database (http://bioinformatics.psb.ugent.be/webtools/plantcare/html/, accessed on 16 August 2022). For example (Appendix A), various cis-acting elements were identified. Photoreaction elements such as Box 4 and the GT1 motif are widely distributed in the promoter regions of *MsPLATZ* genes. In addition to photo-responsive elements, hormone response elements such as ABREs, cell-cycle-related elements such as circadian elements, stress-related elements such as MBSs, and growth and development elements such as CAT boxes are found. Notably, the number of hormone response elements varies greatly among the numerous *MsPLATZ* genes compared to the other elements. As shown in Figure 7, *MsPLATZ42* contains only two auxin-related response elements. In addition to the two auxin-related response elements, *MsPLATZ27* has four elements related to the abscisic acid response, four elements related to the methyl jasmonate response, and one element related to the gibberellin response. These results suggest that different *MsPLATZ* genes may function under different hormonal stimuli.

### 2.7. Differential Expression of MsPLATZ Genes under Different Abiotic Stress Treatments

To understand the responses of the 55 *MsPLATZ* genes involved in abiotic stress responses, we further analyzed the *MsPLATZ* genes that were differentially expressed under three abiotic stresses (salt, cold, and drought). As shown in Figure 8A–C, 20 of the 55 *MsPLATZ* genes show altered expression under the three abiotic stresses, while 3 genes show altered expression under two of the abiotic stresses, among which *MsPLATZ47/37* expression is altered under salt and drought stress, and *MsPLATZ26* expression is altered under cold and drought stress. Four genes (*MsPLATZ48/9/42/1*) are expressed only under one stress, among which, *MsPLATZ48/9* are expressed only under salt stress treatment, *MsPLATZ42* expression is altered only under cold stress, and *MsPLATZ1* is expressed only under drought stress. The expression of *MsPLATZ49/51/54* changes significantly under all three abiotic stress treatments, and the expression abundance of *MsPLATZ49/51/54* also increases significantly with a longer treatment time under both salt and drought treatments. To verify these findings, we performed RT-PCR, the primers are listed in Appendix A. The results show that *MsPLATZ49/51/54* expression significantly increases with prolonged salt and drought stress treatment times (Figure 8D–I). This further indicates that the *MsPLATZ49/51/54* genes may play an important role in stress resistance. Finally, it is concluded that the results obtained for the different genes under the different stress treatments differ significantly, which also indicates that the genes have different functions and specificities. All candidate genes selected in this study are listed in Appendix A.

## 3. Discussion

PLATZ transcription factors are a class of plant-specific zinc-dependent DNA-binding proteins that have been shown by multiple studies to be widely involved in plant growth and development and to play an important role in the response to abiotic stresses [24]. The identification of *PLATZ* gene family members at the genome-wide level has been confirmed in many plants. For example, 12 *PLATZ* genes were identified in *Arabidopsis thaliana* [25], 15 *PLATZ* genes in *Oryza sativa* [25], 17 *PLATZ* genes in *Zea mays* [25], 14 *PLATZ* genes in *Fagopyrum tataricum* [26], 11 *PLATZ* genes in *Ginkgo biloba* [27], and 62 *PLATZ* genes in *Triticum aestivum* [28]. However, systematic analyses and in-depth studies at the genome-wide level are still lacking for *Medicago sativa PLATZ* gene families. In this study, we identified a total of 55 *PLATZ* genes in the alfalfa reference genome, all of which contain a conserved PLATZ domain. Sequence analysis, basic bioinformatics analysis, evolutionary analysis, and expression analysis were also performed. The results indicate that the *MsPLATZ* genes play an important role in the growth and development of *M. sativa* and may cause its resistance to abiotic stresses such as cold, drought, and salinity.

Phylogenetic tree analysis was used to help predict gene function. PLATZ transcription factors regulate root growth in *Arabidopsis thaliana*. For example, in *Arabidopsis thaliana*, *RITF1* (*PLATZ7*), which acts as a receptor of the RGF1 small peptide, is involved in regulating ROS signaling and then in regulating the development of the root meristem [29]. In this study, we find that *MsPLATZ52/47/50/48/46/22/30/26/17/14/55/45/43/44/42* cluster into the same group (group II) as *AtRGFI* (At2G12646, Figure 2). Among these *MsPLATZ* genes, only *MsPLATZ47* is specifically expressed in alfalfa roots (Figure 5). Therefore, through the phylogenetic analysis and expression analysis of the *MsPLATZ* genes, we infer that *MsPLATZ47* may also play an important role in alfalfa root development.

Plants can respond to and coordinate growth and stress resistance by regulating hormone production, distribution, and signaling to promote survival under abiotic and biotic stresses [30]. Zinc finger transcription factors are a relatively large family of plant transcription factors (approximately 15% of the total family) that are involved in regulating the expression of multiple genes in response to cold, salt, drought, osmotic stress, and oxidative stress [31,32]. *AtPLATZ1* and *AtPLATZ12* were identified as major genes positively regulating access to dehydration tolerance in *Arabidopsis thaliana* seeds and vegetative tissues [13]. Transgenic plants overexpressing *AtPLATZ2* show higher sensitivity to salt stress, which inhibits *CBL4/SOS3* promoter activity by directly binding to A/T-rich sequences in the *CBL4/SOS3* and *CBL10/SCaBP8* promoters in vitro and in vivo [24]. In this study, we find 22, 21, and 22 *MsPLATZ* genes responding to cold, drought, and salt stress, respectively (Figure 8A–C). Phylogenetic analysis shows that both *AtPLATZ2* (AT1G76590) and 15 *MsPLATZ* members (*MsPLATZ34/23/31/27/35/38/36/40/37/41/39/54/51/49/53*) belong to group V (Figure 2). Under salt stress, the 11 *MsPLATZ* members are differentially expressed, among which the expression of *MsPLATZ49/51/54* is significantly upregulated (Figure 8D–I). Therefore, these genes may be important candidates for the salt stress response in alfalfa.

Alfalfa is one of the most important feed crops in the world and is known as the “king of forage grass” [20]. It shows high adaptability, high yields, and a nutrient-rich composition and is considered a high-quality diet for herbivores [21]. As alfalfa is often subjected to abiotic stresses such as cold, drought, and salinity during growth, its quality and production decrease [33,34]. Therefore, it is important to identify good alfalfa stress resistance genes to guide alfalfa production. PLATZ transcription factors are zinc finger transcription factors, which constitute a relatively large family of plant transcription factors that are widely involved in plant growth and development and play an important role in the response to abiotic stresses [24]. The results of this study provide useful information for the *MsPLATZ* gene family, and the functions of *MsPLATZs* in response to various abiotic stresses need to be confirmed by more experiments in the future.

## 4. Materials and Methods

### 4.1. Identification of MsPLATZ Genes in the Medicago sativa Genome

The alfalfa genome comes from the Alfalfa Genome Project (https://fgshare.com/projects/whole_genome_sequencing_and_assembly_of_Medicago_sativa/66380, accessed on 5 August 2022). *Arabidopsis thaliana* protein sequences and *Medicago truncatula* genome were obtained from the *Arabidopsis thaliana* Information Resource (TAIR) (https://www.arabidopsis.org/, accessed on 9 August 2022), and (http://www.medicagogenome.org/, accessed on 9 August 2022). The largest number of *PLATZ* genes was screened from the alfalfa genome by BLASTp methods, and the hidden MarKov model (HMM) profiles corresponding to the PLATZ domain (PF04640) were downloaded from the Pfam protein family database (https://pfam.xfam.org/, accessed on 13 August 2022). In total, 55 *MsPLATZ* genes were identified in the alfalfa genome using BLAST with a cutoff E-value > 1e^−10^. All putative *MsPLATZ* genes integrating the results of the HMM retrieval and BLASTP operations were submitted to the NCBI Conserved Domain Database (CDD, https://www.ncbi.nlm.nih.gov/cdd, accessed on 19 August 2022), SMART (http://smart.embl-heidelberg.de/, accessed on 19 August 2022), and Pfam to examine the existence of the conserved PLATZ domain. The ExPASy website (https://web.expasy.org/compute_pi/, accessed on 8 September 2022) was used to analyze the *MsPLATZ* gene sequences to obtain the theoretical isoelectric points (pIs) and molecular weights (MWs).

### 4.2. Phylogenetic Analysis and Intron–Exon Structure Determination

The PLATZ protein sequences used to generate the phylogenetic tree were obtained from the UniProt database (https://www.UniProt.org, accessed on 20 August 2022). The PLATZ protein sequences from *Medicago sativa* and *Arabidopsis thaliana* were also aligned using the Clustalx2.0 program before the phylogenetic tree was constructed. Phylogenetic trees comparing *MsPLATZs* and *AtPLATZs* were constructed with the NJ method, and the specific parameters were the Poisson model and 1000 bootstrap replications in MEGA software (v11.0, Tamura, K., Tokyo, Japan). The classification of *MsPLATZs* based on the phylogenetic tree was performed according to the method described by Wang et al. [25]. The determination of the conserved motifs in the MsPLATZ proteins was conducted by the MEME online program (http:/meme.nbcr.net/meme/intro.html, accessed on 11 August 2022), and the parameters were set to the optimum mode width of 6 to 200 and the maximum number of motifs of 10.

### 4.3. Gene Duplication, Synteny Analysis and Cis-Element Analysis

MCScanX software [35,36] (Hu, Y., Herndon, USA) was used to analyze the *MsPLATZ* replication events and detect collinear regions between *MsPLATZ* and collinear blocks of *MsPLATZ* genes with *Arabidopsis thaliana*, *Medicago truncatula*, and *Glycine max*. All function and chromosomal location information was obtained by TBtools software [37] (v1.108, Chen, C., GZ, China). The upstream 2 kb sequence was extracted as the promoter region for the prediction of cis-acting elements. The homeopathy components of the promoter sequence were predicted by the online tool PlantCARE [38], and the predicted results were drawn by GSDS online software (v2.0, Hu, B., BJ, China).

### 4.4. Plant Materials and Treatments

The alfalfa seeds (Cultivar Zhongmu No. 1) were preserved in our laboratory at the Institute of Animal Science of the Chinese Academy of Agricultural Sciences. The seedlings were placed in a greenhouse at 24 °C (day)/20 °C (night) with a 16 h light/8 h dark photoperiod in the hydroponic culture medium for two weeks. For cold stress, drought stress and salt stress, refer to He et al. [30]. Three replicates with five single seedlings young leaves in each replicate were collected for each condition. The samples were stored at −80 °C for further RT-PCR experiments.

### 4.5. Transcriptome Data Collection and Analysis

Transcriptomic data from six *Medicago sativa* tissues (flower, leaf, elongated stem, pre-elongated stem, nodule, and root) were collected from the NCBI database (SRP055547) [39]. Transcriptomic data for *MsPLATZ* genes exposed to cold, drought, and salt treatments were collected from the NCBI database (SRR7091780–SRR7091794 and SRR7160313– SRR7160357) [40]. TopHat2 was used to map the obtained clean reads to the reference genome (Xinjiangdaye) [41]. In addition, we applied fragments per kilobase of transcript per million fragments mapped (FPKM) to calculate the gene expression level according to the number of reads mapped to the reference sequence. TBtools software (v1.108, Chen, C., GZ, China) was used for data visualization [37].

### 4.6. Expression Analyses of MsPLATZ Genes by qRT-PCR

Total RNA was extracted from all samples in this study by using TRIzol reagent according to the manufacturer’s instructions. Then, the corresponding cDNA was obtained by using the EasyScript First-Strand cDNA Synthesis kit. The RT-PCR primers for these *MsPLATZ* genes were designed by using Primer 5.0 software. The RT-PCR experiments were conducted by using SYBR Premix Ex Taq (Takara, Maebashi, Japan) on a 7500 Real-Time PCR system (Applied Biosystems, Foster City, CA, USA). Three replicates were designed for each sample, and alfalfa actin gene expression was used for data normalization. The data were quantified by the 2^−ΔΔCT^ method [42].

## 5. Conclusions

In this study, the PLATZ gene family members of *Medicago sativa* were systematically identified and characterized. A total of 55 MsPLATZ proteins were identified that were unevenly distributed on 24 out of the 32 chromosomes in *Medicago sativa*, and one was not spliced to the chromosomes. Based on the phylogenetic analysis, the MsPLATZ proteins were divided into five groups, each with similar gene structure and motif compositions. In addition, in regard to gene duplications, gene segment duplication rather than gene tandem duplication is the main driving force of *MsPLATZ* gene evolution. We analyzed the expression levels of 55 *MsPLATZ* genes in different tissues of *Medicago sativa* and their responses to three abiotic stresses (salt, cold, and drought). Expression analysis shows that 7 *MsPLATZ* genes have tissue-specific expression, and 10 *MsPLATZ* genes are expressed in all tissues. The transcriptomic profiles of these genes vary greatly, indicating that these *MsPLATZ* genes have different functions in the growth and development of alfalfa. Through transcriptome data analysis and real-time quantitative PCR (RT-qPCR), we identify 22, 21, and 22 *MsPLATZ* genes that are responsive to cold, drought, and salt stress, respectively. Moreover, the expression of *MsPLATZ49/51/54* under salt and drought stress is significant, indicating that *MsPLATZ49/51/54* may be related to the regulation of salt and drought stress. The results of this study lay a foundation for the further exploration of the *MsPLATZ* function and provide ideas for the improvement of alfalfa varieties and the innovation of germplasm resources.

## Figures and Tables

**Figure 1 ijms-24-02388-f001:**
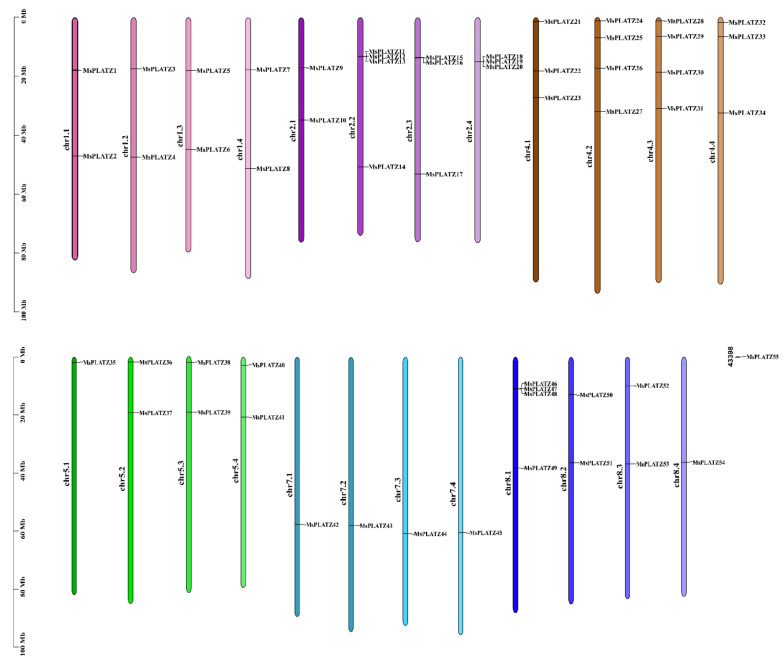
Schematic diagram of the chromosomal distribution of *MsPLATZ* genes in *Medicago sativa*. The vertical bars represent the chromosomes of *Medicago sativa*, and a scale for chromosome length is shown on the left.

**Figure 2 ijms-24-02388-f002:**
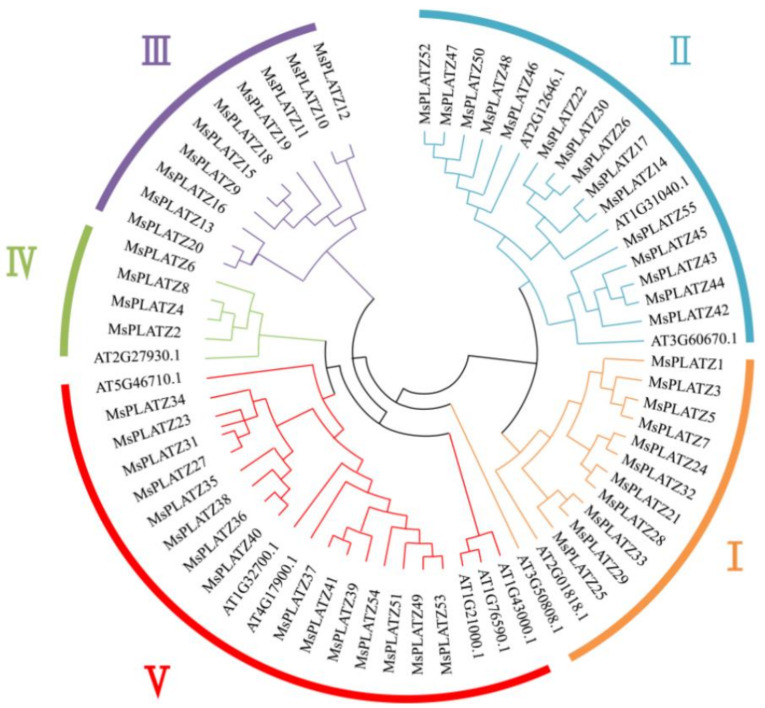
Phylogenetic tree of the *PLATZ* genes in *Medicago sativa* and *Arabidopsis thaliana*. I–V: *PLATZ* genes were divided into five groups and were represented using different colors; Neighbor-joining (NJ) was used to generate the unrooted phylogenetic tree with the maximum likelihood method (1000 bootstrap replicates).

**Figure 3 ijms-24-02388-f003:**
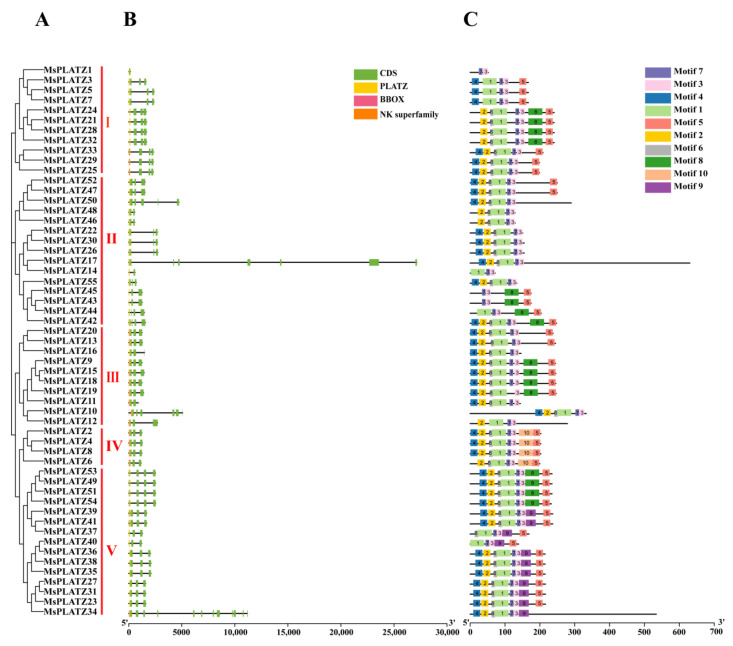
Phylogenetic relationships, gene structures, and motif compositions of *MsPLATZ* genes of *Medicago sativa*. (**A**) The phylogenetic tree of *PLATZ* genes of *Medicago sativa* was constructed using the maximum likelihood method. I–V: *MsPLATZ* genes were divided into five groups. (**B**) Exon–intron structure of *MsPLATZ* genes. The legend is shown in the upper-right corner, and the introns are represented by black lines. (**C**) Motif composition of *MsPLATZ* genes. Different motifs are represented by different colors, as indicated in the legend on the right.

**Figure 4 ijms-24-02388-f004:**
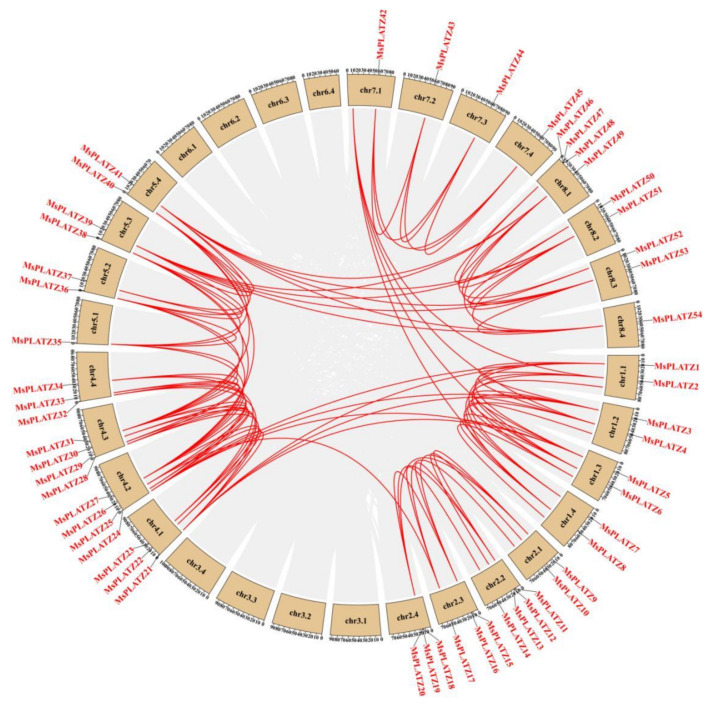
Schematic diagram of the syntenic relationships of *MsPLATZ* genes in *Medicago sativa*. The gray ribbons represent syntenic blocks in the alfalfa genome, and the segmental duplication events are marked in red.

**Figure 5 ijms-24-02388-f005:**
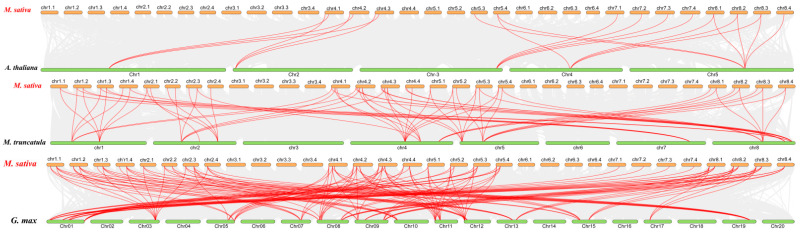
Synteny analysis of *PLATZ* genes between *Medicago sativa* and three representative plant species. Gray lines in the background indicate collinear blocks within *Medicago sativa* and the indicated plant species, whereas the red lines highlight syntenic *PLATZ* gene pairs.

**Figure 6 ijms-24-02388-f006:**
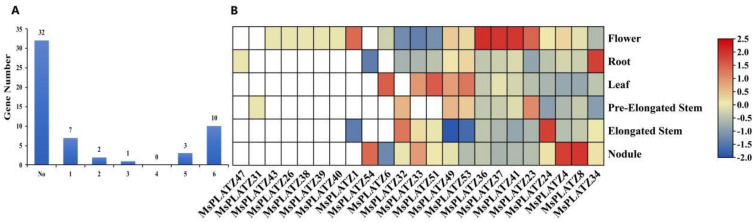
Expression analysis of *MsPLATZ* genes in different tissues (flower, leaf, elongated stem, pre−elongated stem, nodule, and root). (**A**) Statistical analysis of the numbers of *MsPLATZ* genes expressed in different tissues. (**B**) Expression of 23 *MsPLATZ* genes in different tissues. The expression levels were normalized by row using the Z−scores algorithm. The color scale on the right side of the heatmap indicates the relative expression levels, and the color gradient from blue to red indicates an increase in expression levels.

**Figure 7 ijms-24-02388-f007:**
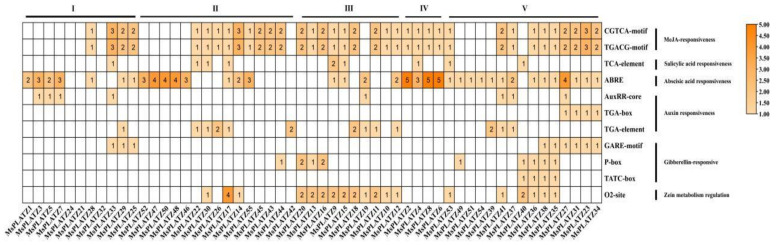
Distribution of the cis-acting elements related to hormone responses in the promoter regions of *MsPLATZ* genes. I–V: *MsPLATZ* genes were divided into five groups.

**Figure 8 ijms-24-02388-f008:**
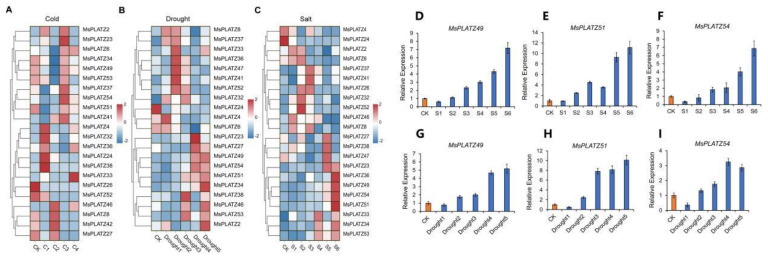
Expression of *MsPLATZ* genes under different abiotic stress treatments (salt, cold, drought). (**A**) Expression of *MsPLATZ* genes under cold stress treatments. (**B**) Expression of *MsPLATZ* genes under drought stress treatments. (**C**) Expression of *MsPLATZ* genes under salt stress treatments. The expression levels were normalized by row using the Z−scores algorithm. The color scale at the right of the heatmap refers to the relative expression level, and the color gradient from blue to red presents an increasing expression level. (**D**–**F**). RT-PCR expression of *MsPLATZ49/51/54* under salt stress. (**G**–**I**). RT-PCR expression of *MsPLATZ49/51/54* under drought stress. CK was set to orange, and the treatment groups were set to blue. CK was arbitrarily set to 1. Error bars represent the standard deviations of three technical replicates.

**Table 1 ijms-24-02388-t001:** *PLATZ* family genes in *Medicago sativa*.

Gene Name	Gene ID	Chr Location	CDS Length (bp)	Protein Length (aa)	Mw (kDa)	pI	Subcellular Location
*MsPLATZ1*	MS.gene026494	Ms1.1: 17,863,747–17,863,902	156	52	5.83	9.64	Nucleus
*MsPLATZ2*	MS.gene051384	Ms1.1: 46,909,227–46,910,461	609	203	22.92	8.83	Nucleus
*MsPLATZ3*	MS.gene040928	Ms1.2: 17,427,194–17,428,828	501	167	19.37	8.97	Nucleus
*MsPLATZ4*	MS.gene029705	Ms1.2: 47,307,863–47,309,143	609	203	22.95	8.83	Nucleus
*MsPLATZ5*	MS.gene95330	Ms1.3: 17,964,734–17,967,118	501	167	19.36	8.97	Nucleus
*MsPLATZ6*	MS.gene034971	Ms1.3: 44,693,845–44,695,016	600	200	22.61	8.96	Nucleus
*MsPLATZ7*	MS.gene83024	Ms1.4: 17,595,429–17,597,805	501	167	19.36	8.97	Nucleus
*MsPLATZ8*	MS.gene47395	Ms1.4: 51,168,875–51,170,109	609	203	22.92	8.7	Nucleus
*MsPLATZ9*	MS.gene35330	Ms2.1: 17,012,467–17,013,708	735	245	28.58	9.24	Nucleus
*MsPLATZ10*	MS.gene99752	Ms2.1: 34,752,509–34,757,591	999	333	36.91	4.69	Chloroplast
*MsPLATZ11*	MS.gene073125	Ms2.2: 13,169,890–13,170,791	432	144	16.70	9.03	Cytosol
*MsPLATZ12*	MS.gene073126	Ms2.2: 13,185,522–13,188,245	837	279	31.60	5.81	Extracellular
*MsPLATZ13*	MS.gene073129	Ms2.2: 13,219,049–13,220,333	735	245	28.46	8.71	Extracellular
*MsPLATZ14*	MS.gene57445	Ms2.2: 50,631,783–50,632,379	216	72	8.39	8.85	Chloroplast
*MsPLATZ15*	MS.gene91274	Ms2.3: 13,498,110–13,499,564	738	246	28.68	9.26	Nucleus
*MsPLATZ16*	MS.gene91270	Ms2.3: 13,545,547–13,547,034	438	146	16.93	9.21	Extracellular
*MsPLATZ17*	MS.gene58706	Ms2.3: 53,047,725–53,074,883	1890	630	72.48	9	Nucleus
*MsPLATZ18*	MS.gene94621	Ms2.4: 14,906,505–14,907,746	738	246	28.71	9.26	Nucleus
*MsPLATZ19*	MS.gene94620	Ms2.4: 14,914,416–14,915,814	741	247	28.69	9.1	Nucleus
*MsPLATZ20*	MS.gene94618	Ms2.4: 14,941,107–14,942,368	714	238	27.88	8.79	Extracellular
*MsPLATZ21*	MS.gene015620	Ms4.1: 1,317,444–1,319,118	723	241	27.86	8.94	Nucleus
*MsPLATZ22*	MS.gene36510	Ms4.1: 18,036,419–18,039,121	453	151	16.90	8.57	Nucleus
*MsPLATZ23*	MS.gene065915	Ms4.1: 27,206,398–27,208,000	648	216	23.96	9.31	Nucleus
*MsPLATZ24*	MS.gene62657	Ms4.2: 1,067,724–1,069,373	723	241	27.90	9.03	Nucleus
*MsPLATZ25*	MS.gene72172	Ms4.2: 6,826,199–6,828,514	597	199	22.67	8.82	Nucleus
*MsPLATZ26*	MS.gene028280	Ms4.2: 17,145,854–17,148,601	465	155	17.48	8.57	Nucleus
*MsPLATZ27*	MS.gene83177	Ms4.2: 31,843,428–31,845,025	648	216	23.96	9.31	Nucleus
*MsPLATZ28*	MS.gene95852	Ms4.3: 1,135,074–1,136,748	723	241	27.86	8.94	Nucleus
*MsPLATZ29*	MS.gene048775	Ms4.3: 6,372,228–6,374,549	597	199	22.67	8.82	Nucleus
*MsPLATZ30*	MS.gene023191	Ms4.3: 18,576,051–18,578,761	465	155	17.40	8.57	Nucleus
*MsPLATZ31*	MS.gene031722	Ms4.3: 30,835,572–30,837,171	648	216	23.96	9.31	Nucleus
*MsPLATZ32*	MS.gene058564	Ms4.4: 1,743,997–1,745,674	723	241	27.89	9.03	Nucleus
*MsPLATZ33*	MS.gene030429	Ms4.4: 6,545,886–6,548,219	627	209	23.65	8.81	Nucleus
*MsPLATZ34*	MS.gene006587	Ms4.4: 32,411,516–32,422,710	1602	534	58.31	8.69	Nucleus
*MsPLATZ35*	MS.gene058083	Ms5.1: 1,754,805–1,756,904	645	215	24.06	9.07	Nucleus
*MsPLATZ36*	MS.gene21784	Ms5.2: 1,651,210–1,653,257	645	215	24.16	9.15	Nucleus
*MsPLATZ37*	MS.gene64179	Ms5.2: 18,922,507–18,923,800	507	169	19.24	9.91	Nucleus
*MsPLATZ38*	MS.gene40512	Ms5.3: 2,100,515–2,102,615	645	215	24.06	9.07	Nucleus
*MsPLATZ39*	MS.gene036303	Ms5.3: 19,070,783–19,072,470	711	237	27.11	9.44	Nucleus
*MsPLATZ40*	MS.gene024806	Ms5.4: 2,681,147–2,682,369	417	139	15.60	9.72	Nucleus
*MsPLATZ41*	MS.gene027893	Ms5.4: 20,364,134–20,365,827	711	237	27.11	9.44	Nucleus
*MsPLATZ42*	MS.gene96355	Ms7.1: 56,837,237–56,838,808	744	248	27.69	8.99	Extracellular
*MsPLATZ43*	MS.gene20118	Ms7.2: 57,201,793–57,203,057	525	175	18.96	9.22	Nucleus
*MsPLATZ44*	MS.gene82045	Ms7.3: 59,965,702–59,967,169	612	204	22.34	9.31	Extracellular
*MsPLATZ45*	MS.gene010738	Ms7.4: 59,599,942–59,601,202	525	175	18.93	9.22	Nucleus
*MsPLATZ46*	MS.gene036665	Ms8.1: 10,732,368–10,732,928	387	129	15.08	8.84	Cytosol
*MsPLATZ47*	MS.gene59920	Ms8.1: 10,748,163–10,749,697	750	250	28.74	8.61	Nucleus
*MsPLATZ48*	MS.gene041385	Ms8.1: 10,830,973–10,831,533	387	129	15.08	8.84	Cytosol
*MsPLATZ49*	MS.gene60992	Ms8.1: 37,680,744–37,683,265	705	235	26.73	9.5	Nucleus
*MsPLATZ50*	MS.gene041387	Ms8.2: 12,789,586–12,794,323	870	290	33.82	8.67	Nucleus
*MsPLATZ51*	MS.gene009009	Ms8.2: 35,918,983–35,921,509	705	235	26.74	9.5	Nucleus
*MsPLATZ52*	MS.gene58786	Ms8.3: 9,688,042–9,689,586	750	250	28.74	8.61	Nucleus
*MsPLATZ53*	MS.gene71193	Ms8.3: 36,233,879–36,236,397	705	235	26.73	9.5	Nucleus
*MsPLATZ54*	MS.gene038444	Ms8.4: 35,728,371–35,730,908	699	233	26.50	9.5	Nucleus
*MsPLATZ55*	MS.gene81453	43398: 41,531–42,243	402	134	15.69	9.1	Nucleus

Chr: chromosome; CDS: coding sequence; bp: base pair; aa: amino acid; Mw: molecular weight; pI: isoelectric point.

## Data Availability

Not applicable.

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
