# Peer review of "Genome-Wide Identification and Phylogenetic and Expression Analyses of the PLATZ Gene Family in Medicago sativa L."

_ijms, 2023, doi:10.3390/ijms24032388_

Round 1
Reviewer 1 Report
This MS is very well prepared and requires only a minor spell check. The authors did use the word 'specific' 4 times in the first sentence of the Introduction:). This reviewer can find no issues with the quality of the MS or its presentation or the data analysis and congratulate the authors on their accomplishment.
Reviewer 2 Report
Dear authors, I think that your work is good, scientifically significant, the MS is well written. I recommend its publication in the journal.
However, I recommend three minor technical corrections.
Line 82 is known as the "King of forage grasses" [20-21]..
Line 327 Alfalfa is one of the most important feed crops in the world and is known as the "king of forage grass"[20].
I suppose it will suffice to mention this metaphor once.Lines 114-116 The greatest number of MsPLATZ genes (four genes) were located on chromosomes chr2.2, chr4.2, chr4.3, and chr8.1, while three genes were distributed on chromosomes chr2.3, chr2.4, chr4.1, and chr4.3. In chr1.1, chr1.2, chr1.3, chr1.4, chr2.1, chr2.2, chr5.2, chr5.3, 117 chr5.4, chr8.2, and chr8.3, two genes were distributed on these 10 chromosomes
Chromosomes 2.2 and 4.3 are listed twice by mistake.Please check the use of italics or roman type
in the names of genes and proteins, for example, in lines 62, 108, 128, 131, 133, 171 428. A PLATZ protein plays an important role in the 62 of the MsPLATZ genes were quite different, 108
between the MsPLATZ proteins and the 128 PLATZ proteins were divided into five groups (I - V), and
the MsPLATZ proteins were 131
group â…¤ contained the most PLATZ proteins 132-133 components of MsPLATZ proteins. 171 that these MsPLATZ genes have different functions in the growth and development of 428
Reviewer 3 Report
The present study pivots around identification and characterization of the PLATZ members which have been identified in many crop plants, but due to the complexity of the alfalfa reference genome, the members of the PLATZ gene family in alfalfa have not been systematically studied. In a large way basic bioinformatic analysis was performed to reveal the intricacies PLATZ members at sequence and protein level. Appreciable results obtained from the expression analysis by identifying the 7 MsPLATZ genes which had tissue-specific expression, and 10 MsPLATZ genes which were expressed in all the tissues. Overall the experimental set-up, measurements, bioinformatics analysis and statistical treatment seem to be carefully planned and conducted and the work is original and novel.
Following are the suggestions for the betterment of the article :
1. For each bioinformatics analysis more details to be added so that the results can be reproducible. How the sequence search was performed by using hidden Markova Model? Similarly more details can be added in the Phylogenetic analysis and the gene duplication and synteny analysis.
2. While using leaves from the 2 weeks old seedlings whether young leaves were preferred or the older leaves? Any criteria were used for selecting the leaves?
3. Any interesting results in comparison from the downloaded transcriptomic data for genes exposed to ABA treatments?
4. List out the various candidate MsPLATZ genes involved in various abiotic stress responses in alfalfa for particular stress in the separate table.
Hence, my final decision on the article is to accept with minor revision.
